# GEM: 3D GAUSSIAN SPLATTING FOR EFFICIENT AND ACCURATE CRYO-EM RECONSTRUCTION

## ABSTRACT

Cryo-electron microscopy (cryo-EM) has become a central tool for high-resolution structural biology, yet the massive scale of datasets (often exceeding 100k particle images) renders 3D reconstruction both computationally expensive and memory intensive. Traditional Fourier-space methods are efficient but lose fidelity due to repeated transforms, while recent real-space approaches based on neural radiance fields (NeRFs) improve accuracy but incur cubic memory and computation overhead. Therefore, we introduce GEM, a novel cryo-EM reconstruction framework built on 3D Gaussian Splatting (3DGS) that operates directly in real-space while maintaining high efficiency. Instead of modeling the entire density volume, GEM represents proteins with compact 3D Gaussians, each parameterized by only 11 values. To further improve the training efficiency, we designed a novel gradient computation to 3D Gaussians that contribute to each voxel. This design substantially reduced both memory footprint and training cost. On standard cryo-EM benchmarks, GEM achieves up to $48\times$ faster training and $12\times$ lower memory usage compared to state-of-the-art methods, while improving local resolution by as much as $38.8\%$. These results establish GEM as a practical and scalable paradigm for cryo-EM reconstruction, unifying speed, efficiency, and high-resolution accuracy.

## 1 INTRODUCTION

Cryo-electron microscopy (cryo-EM) (Bai et al., 2015a; Milne et al., 2013b) is a transformative biological imaging technique that enables the determination of macromolecular structures at near-atomic resolution, and its significance was recognized with the 2017 Nobel Prize in Chemistry (Pre). In cryo-EM, biological samples are rapidly frozen in vitreous ice and imaged under an electron microscope, producing thousands to millions of noisy two-dimensional projection images of individual particles. Each projection corresponds to a different orientation of the same underlying macromolecule. By aggregating these projections, the three-dimensional electron density of the protein can be reconstructed (Murata & Wolf, 2018). The cryo-EM imaging model naturally aligns with the Fourier slice theorem, where each projection corresponds to a central slice of the 3D Fourier transform of the density, enabling efficient reconstruction in Fourier space (Zhong et al., 2021; Punjani et al., 2017; Scheres, 2012; Tang et al., 2007).

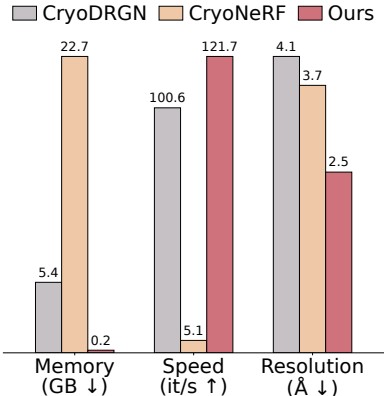

Figure 1: GEM achieves lower memory usage, faster speed, and higher reconstruction resolution compared to existing approaches.

Despite its efficiency, the reliance on Fourier transforms introduces information loss in the traditional methods, limiting achievable resolution. Recent works propose adapting neural radiance fields (NeRFs) to cryo-EM by modeling protein densities directly in real-space (Huang et al., 2024; Qu et al., 2025; Liu et al., 2023), using representations such as MLPs (Liu et al., 2023), Transformers (Huang et al., 2024), or multi-resolution hash encoding (Qu et al., 2025). These approaches remove the dependence on Fourier transforms in the Fourier slice theorem (see Appendix 6.1 for more details) and demonstrate improved resolution compared to traditional Fourier-space pipelines. However, direct application of NeRF to cryo-EM remains impractical. Unlike view synthesis tasks, cryo-EM requires reconstruction of the full 3D density

rather than rendering sparse pixels. And each training step involves applying the contrast transfer function (CTF) across entire projection images, resulting in cubic scaling of memory and computation. Even for small proteins, this requires millions of sampled points and gradients, making NeRF-based cryo-EM reconstruction prohibitively slow and memory-intensive, especially on commodity GPUs.

To address these challenges, we propose GEM, a novel framework that introduces 3D Gaussian Splatting (3DGS) (Kerbl et al., 2023; Zha et al., 2024) for cryo-EM reconstruction. Unlike NeRF-based approaches that model density implicitly, GEM adopts an explicit representation using a set of 3D Gaussians, each parameterized by only 11 values. This representation is both compact and efficient: ($i$) 3D Gaussians are instantiated only at nonzero-density regions, and during projection, ($ii$) each 3D Gaussian contributes to multiple pixels and can be rasterized in parallel. Furthermore, ($iii$) gradient computation is restricted to the subset of Gaussians influencing a given pixel, in contrast to NeRF methods that require activating the entire network for each pixel. This design eliminates the cubic memory footprint of NeRFs and yields substantial acceleration in training.

We validate GEM on four widely used cryo-EM datasets under diverse evaluation protocols. As shown in Figure 1, GEM achieves substantially faster training and lower memory usage than the real-space NeRF-based pipeline CryoNeRF, while also surpassing both the Fourier-space baseline CryoDRGN and the real-space baseline CryoNeRF in reconstruction quality. Across all datasets, GEM consistently attains gold-standard Fourier shell correlation (GSFSC) resolutions better than 3 Å, outperforming existing methods. Additional evaluations further confirm its resolution advantage, and compared to CryoNeRF (Qu et al., 2025), GEM not only improves quality but also delivers significantly higher efficiency, even surpassing the Fourier-based CryoDRGN.

Our main contributions are summarized as follows:

- We introduce GEM, a novel framework that incorporates 3D Gaussian Splatting into cryo-EM reconstruction, providing an explicit, efficient, and accurate representation of protein density that avoids the information loss of Fourier-based methods and the cubic overhead of NeRF-based real-space approaches.

- By explicitly modeling protein density with 3D Gaussians placed only in nonzero-density regions and enabling parallel rasterization across pixels, GEM achieves up to $48\times$ faster training speed compared to existing methods.

- We design an efficient gradient computation strategy that restricts updates to only the 3D Gaussians contributing to each pixel, thereby eliminating the cubic memory footprint and computation overhead of prior real-space methods.

- Extensive experiments on four widely used cryo-EM datasets under diverse evaluation protocols demonstrate that GEM consistently outperforms existing approaches, often reaching resolutions close to the instrumental limitation.

## 2 RELATED WORK

**Cryo-EM reconstruction.** Classical single-particle cryo-EM pipelines formulate reconstruction in Fourier space via the Fourier slice theorem, solving for the 3D density with iterative refinement (Zhong et al., 2021; Punjani et al., 2017; Scheres, 2012; Tang et al., 2007). This formulation enables efficient global solvers and robust regularization schemes and has underpinned much of the field's progress (Milne et al., 2013a; Murata & Wolf, 2018; Bai et al., 2015b). Quality is typically assessed using gold-standard Fourier shell correlation (GSFSC) (Van Heel & Schatz, 2005), local resolution (Glaeser et al., 2021) and directional consistency analyses (Huang et al., 2024). Despite their maturity, Fourier-space methods involve repeated FFTs and interpolation steps and operate on band-limited representations, which can incur information loss. Our work targets these pain points by adopting 3D Gaussian Splatting as an explicit, efficient real-space representation while remaining compatible with standard cryo-EM pipelines and evaluation protocols.

**Real-space cryo-EM reconstruction.** A growing line of work reconstructs density directly in Euclidean space using neural implicit fields and related parameterizations. Neural-field approaches, which often use MLPs with Fourier or hash encodings, or Transformer-based fields, optimize a continuous density so that its simulated projections match observed images, thereby avoiding

frequency-domain interpolation and enabling real-space regularization (Huang et al., 2024; Qu et al., 2025; Liu et al., 2023; Herreros et al., 2025; Palmer & Aylett, 2022; Lu et al., 2022). These methods have demonstrated improved resolution on several benchmarks. However, directly training neural fields for cryo-EM is computationally demanding: unlike view synthesis, we must reconstruct full volumes rather than sparse rays, apply CTFs per micrograph, and backpropagate through dense sampling, leading to memory/time that scale roughly with the volume (and number of samples) instead of the number of visible pixels. This cubic-like footprint makes large proteins and commodity GPUs challenging and motivates representations whose gradients remain localized to the subset of primitives that actually contribute to each pixel.

**3D Gaussian splatting.** 3D Gaussian Splatting (3DGS) (Kerbl et al., 2023; Zha et al., 2024; Wu et al., 2024; Fei et al., 2024; Yu et al., 2024; Ye et al., 2025) represents scenes as explicit sets of anisotropic Gaussians rendered by fast rasterization, yielding state-of-the-art real-time view synthesis with memory that scales with the number of Gaussians rather than the discretized volume (Kerbl et al., 2023). For biological settings, recent work proposes rectified radiative Gaussian splatting to improve fidelity under line/slice integrals (Zha et al., 2024; 2022; Yu et al., 2025), suggesting that splatting is a promising alternative to implicit fields when projections are the fundamental measurements. Adapting 3DGS to cryo-EM introduces two key advantages: ($i$) sparsity, since Gaussians need only occupy non-empty regions of the macromolecule, and ($ii$) locality, since gradients for a pixel involve only the Gaussians that cover it, avoiding the cubic memory of dense neural fields. Our framework instantiates these ideas for single-particle cryo-EM by coupling a CTF-aware, differentiable projection of Gaussians with joint optimization over density and imaging latents.

## 3 METHOD

### 3.1 CRYO-EM RECONSTRUCTION

Cryo-EM reconstruction (Elmlund & Elmlund, 2015) aims to recover the 3D structure of a protein from a large collection of noisy 2D particle images. In practice, each image is acquired as a projection of a protein particle illuminated by an electron beam, modulated by the microscope's contrast transfer function (CTF) and corrupted by background noise.

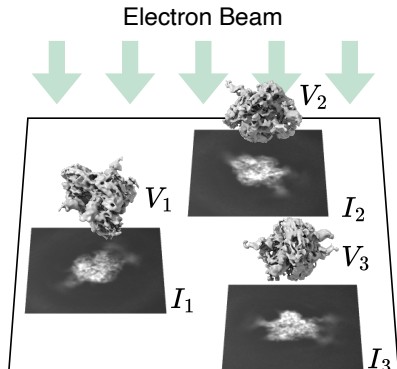

Figure 2: Cryo-EM imaging process. Randomly oriented proteins are illuminated by the electron beam, leaving traces on the sensor as particle images $I_1$, $I_2$, and $I_3$. In this work we focus on homogeneous reconstruction, where all particles $V_1$, $V_2$, and $V_3$ share the same underlying 3D structure and differ only in orientation (i.e., multiple copies of the same object).

**Image Formation Model.** Let $\mathcal{I} = \{I_i\}_{i=1}^N$ denote a dataset of $N$ particle images and $\mathcal{V} = \{V_i\}_{i=1}^N$ the corresponding particles. Each image $I_i \in \mathbb{R}^{d \times d}$ in Figure 3 is a projection of an underlying 3D density map $V_i \in \mathbb{R}^{d \times d \times d}$ following the imaging model that can be expressed as

$$I_i = C_i * \mathrm{Proj}\left(V_i; \phi_i, t_i\right) + \eta_i, \tag{1}$$

where $C_i$ is the CTF determined by microscope settings and $*$ is the convolution operation. $\mathrm{Proj}(V) = \int_{-\infty}^0 V dz$ denotes the projection operator with rotation angles $\phi_i$ and translation $t_i$, and $\eta_i$ represents additive noise.

**Homogeneous Reconstruction.** In this work, we focus on homogeneous reconstruction, where all particles, e.g. $V_1$, $V_2$, and $V_3$ in Figure 3, share the same underlying structure:

$$V_i = V, \ \forall \ V_i \in \mathcal{V}. \tag{2}$$

The goal is then to recover the shared 3D density $V$ from all images and their associated parameters:

$$\hat{V} = \mathrm{Reconstruct}\left(\mathcal{I}; \Phi, \mathcal{T}, \mathcal{C}\right), \tag{3}$$

where $\Phi = \{\phi_i\}_{i=1}^N$, $\mathcal{T} = \{t_i\}_{i=1}^N$, and $\mathcal{C} = \{C_i\}_{i=1}^N$ denote the set of rotations, translations and CTFs, which are known parameters before the reconstruction.

**Fourier Slice Theorem for Reconstruction:** Fourier-space methods leverage the Fourier slice theorem and therefore only need to predict a slice in the Fourier domain, i.e., $\mathrm{Slice}(\mathcal{F}(\hat{V}_i); , \phi_i, t_i)$, without instantiating the full protein density $\hat{V}$. **Real-Space Cryo-EM Reconstruction:** In contrast, real-space methods instantiate the entire volume $\hat{V}$ and perform projection following Equation 1, which requires cubic memory to store both the predicted values and their gradients.

## 3.2 3D GAUSSIAN SPLATTING FOR RECONSTRUCTION

We present the GEM training framework in Figure 4. First, protein density can be represented as a set of 3D Gaussians (Figure 3a). To optimize the randomly initialized 3D Gaussian-based representation, we can project 3D Gaussian representations to 2D space. The 3D Gaussian representations can then be optimized by minimizing the differences between the projected 2D images and the captured 2D images by the real microscope. During training, we also introduced novel efficient gradient computation to reduce memory cost. After training, the reconstructed protein density can be queried from the trained 3D Gaussians (Figure 3b).

**Protein Density represented by 3D Gaussians.** Following previous 3DGS literature (Kerbl et al., 2023; Zha et al., 2024), we model the protein density as a set of 3D Gaussians $\mathcal{G} = \{G_j\}_{j=1}^{M}$, where each Gaussian is parameterized by its center $\mathbf{p}_j$, covariance matrix $\boldsymbol{\Sigma}_j$, and density coefficient $\rho_j$. Each kernel $G_j$ defines a local Gaussian-shaped density field:

$$G_j(\mathbf{x} \,|\, \rho_j, \mathbf{p}_j, \boldsymbol{\Sigma}_j) = \rho_j \cdot \exp\left(-\frac{1}{2}(\mathbf{x} - \mathbf{p}_j)^\top \boldsymbol{\Sigma}_j^{-1}(\mathbf{x} - \mathbf{p}_j)\right), \qquad \boldsymbol{\Sigma}_j = \mathbf{R}_j \mathbf{S}_j \mathbf{S}_j^\top \mathbf{R}_j^\top. \quad (4)$$

where the covariance $\boldsymbol{\Sigma}_j$ is further decomposed into a rotation $\mathbf{R}_j$ and a scale matrix $\mathbf{S}_j$. The overall density of protein $V_i$ at position $\mathbf{x}$ is then given by the summation of all Gaussian kernels:

$$\hat{V}_i(\mathbf{x}) = \sum_{j=1}^{M} G_j(\mathbf{x} \,|\, \rho_j, \mathbf{p}_j, \boldsymbol{\Sigma}_j). \quad (5)$$

Compared to the original 3DGS formulation, which incorporates view-dependent effects for neural rendering, our variant adopts an isotropic, density-based, density-based formulation consistent with the cryo-EM imaging model, where reconstruction relies on where reconstruction relies on direct density integration along the electron beam.

**Fast 2D Projection of 3D Gaussian representations via Explicit Integration for .** Owing to the property of 3D Gaussians, the projection can be expressed in a closed form, avoiding querying density values and then aggregating them. Let $\hat{V}_i(\mathbf{x})$ denote the protein density for view $i$. The volume is defined as a sum of 3D 3D Gaussians following Equation 5, and the integration along the optical axis $z$ yields a 2D Gaussian mixture:

$$\hat{I}_i(\mathbf{r}) = \sum_{j=1}^{M} \rho_j \frac{1}{2\pi \left|\hat{\boldsymbol{\Sigma}}_j\right|^{1/2}} \exp\left(-\frac{1}{2}(\hat{\mathbf{x}} - \hat{\mathbf{p}}_j)^\top \hat{\boldsymbol{\Sigma}}_j^{-1}(\hat{\mathbf{x}} - \hat{\mathbf{p}}_j)\right) = \sum_{j=1}^{M} G_j^2, \quad (6)$$

where $\hat{\mathbf{x}} \in \mathbb{R}^2$, $\hat{\mathbf{p}}_j \in \mathbb{R}^2$, and $\hat{\boldsymbol{\Sigma}}_j \in \mathbb{R}^{2 \times 2}$ are the in-plane (mean, covariance) obtained by dropping the third row/column of their 3D counterparts $\mathbf{x}$, $\mathbf{p}_j$, and $\boldsymbol{\Sigma}_j$, respectively. This closed-form marginalization enables efficient CUDA kernels and removes the need to explicitly query $\hat{V}_i$ on 3D grids before the projection, substantially reducing both compute and memory. Please see Appendix 6.2 for more detail of the derivation of the projection. As shown in Figure 4, Step1 applies Equation 6 to generate the projection from the protein density.

**Optimizing 3D GS representations** Given the projected image $\hat{I}_i$, we apply the CTF $C_i$ in Fourier space and compare to the noisy measurement $I_i$ via an $\ell_2$ loss:

$$I_i^{\mathrm{pred}} = \mathcal{F}^{-1}\big(\mathcal{F}(C_i) \cdot \mathcal{F}(\hat{I}_i)\big), \qquad \mathcal{L}_i = \left\|I_i^{\mathrm{pred}} - I_i\right\|_2^2, \quad (7)$$

where $\mathcal{F}$ and $\mathcal{F}^{-1}$ are Fourier and inverse Fourier transforms, respectively. In Equation 7, Fourier transform is utilized as the convolution in Equation 1 equals to the multiplication in Fourier space

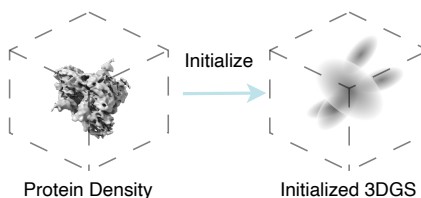 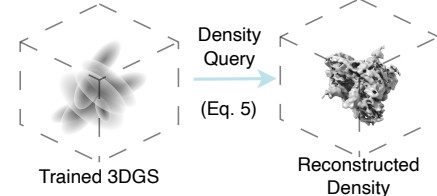

(a) Protein density is represented by 3D Gaussians. Before training, the parameters of 3D Gaussians are randomly initialized.

(b) After training, the density reconstruction is queried from the trained 3D Gaussians following Equation 5.

Figure 3: Initialization before training and density reconstruction after training.

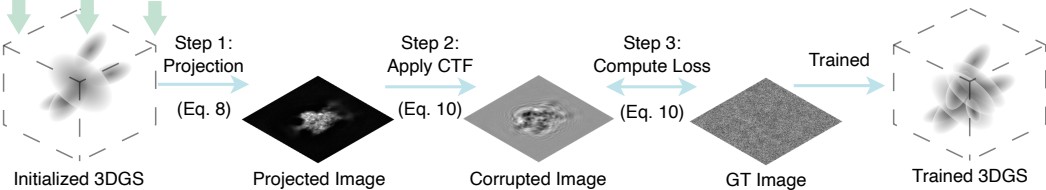

Figure 4: The overview of GEM training. The training begins with randomly initialized Gaussians. The 3D Gaussians are projected following Equation 6. Then the CTF is applied (Equation 7) to the projection and being compared with the noisy experimental image to calculate the loss (Equation 7).

(McGillem & Cooper, 1991). Although 3DGS is typically trained on clean natural images, we observe that the standard MSE loss is sufficient for noisy cryo-EM images. Equation 7 corresponds to Step 2 and 3 in Figure 4, where the CTF is first applied to the projected image, followed by computing the standard MSE loss between the CTF-corrected image and the experimental ground-truth image.

**Efficient Gradient Computation to Reduce Memory Overhead.** While our rasterization implementation follows the principle of Equation 6, directly evaluating all 3D Gaussians remains inefficient. To address this, we introduce a thresholding strategy that selects only the Gaussians with non-negligible contributions to a given ray $\mathbf{r}$. Formally,

$$\hat{I}_i(\mathbf{r}) = \sum_{j=1}^{K} G_j^2, \quad \text{where } G_j^2 > \tau, \tag{8}$$

where $\tau$ is a predefined threshold. This pruning avoids unnecessary computation from Gaussians far from the ray, since their contribution decays quadratically with distance to the Gaussian center $\mathbf{p}_j$.

After selection, the remaining 3D Gaussians are sorted along the $z$-axis, and the accumulation in Equation 8 is performed from the lowest $z$-value (closest to the cryo-EM sensor) to the highest (farthest from the sensor). This ordering prioritizes the most influential 3D Gaussians during rendering and further improves computational efficiency.

## 4 EXPERIMENT

In this section, we evaluate the proposed method GEM using four widely adopted cryo-EM datasets: EMPIAR-10005 (TRPV1, Liao et al. (2013)), EMPIAR-10028 (*Plasmodium falciparum* 80S ribosome, Wong et al. (2014)), EMPIAR-10049 (Synaptic RAG1-RAG2 complex, Ru et al. (2015)), and EMPIAR-10076 (L17-depleted 50S ribosomal intermediates, Davis et al. (2016)).

To demonstrate the effectiveness of GEM, we compare it with two representative cryo-EM reconstruction approaches: (i) the conventional voxel-based method CryoSPARC (Punjani et al., 2017), and (ii) the neural network-based CryoDRGN (Zhong et al., 2021). Both of the methods reconstruct cryo-EM densities in Fourier space levaraging the Fourier slice theorem (Radon, 2005; Garces et al., 2011), in contrast to our GEM that operates in 3D Euclidean space. For CryoSPARC, we first perform *ab initio*

Table 1: Efficiency comparison of deep learning-based cryo-EM reconstruction methods. Speed (it/s) measures the number of images processed per second, an Memor (GB) reports the maximum GPU memory consumption during training. All experiments are conducted on a single NVIDIA RTX A6000 (48 GB). OOM indicates out-of-memory errors. **Our method achieves the highest training throughput while requiring substantially less memory across all datasets.**

|  | EMPIAR-10028 | | EMPIAR-10049 | | EMPIAR-10076 | | EMPIAR-10005 | |
|---|---|---|---|---|---|---|---|---|
|  | Speed (it/s) ↑ | Memory (GB) ↓ | Speed (it/s) ↑ | Memory (GB) ↓ | Speed (it/s) ↑ | Memory (GB) ↓ | Speed (it/s) ↑ | Memory (GB) ↓ |
| CryoDRGN | 47.62 | 8.81 | 142.86 | 2.58 | 58.82 | 8.79 | 100.00 | 4.93 |
| CryoNeRF | - | OOM | 9.13 | 9.94 | 2.10 | 37.75 | 4.03 | 20.50 |
| Ours | **94.10** | **1.54** | **160.49** | **0.50** | **96.60** | **0.74** | **108.11** | **0.51** |

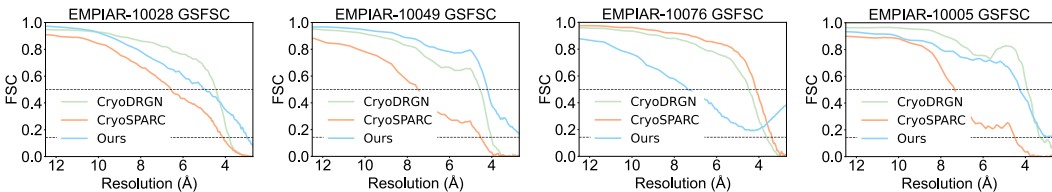

Figure 5: GSFSC of our `GEM` and baselines. The horizontal axis denotes resolution in ångströms (Å), and the vertical axis denotes the FSC value. The two dashed horizontal lines are FSC thresholds of 0.5 and 0.143. The final resolution is defined as the resolution at which the GSFSC curve first drops below the 0.143 threshold (smallest possible resolution if no intersection). **Intersection points further to the right corresponds to better reconstruction quality.**

Table 2: Final resolutions (in Å) obtained by each method according to the GSFSC 0.143 threshold. **Lower values indicate better resolution.**

|  | EMPIAR-10028 | EMPIAR-10049 | EMPIAR-10076 | EMPIAR-10005 |
|---|---|---|---|---|
| CryoSPARC | 4.12 Å | 4.53 Å | 3.39 Å | 4.50 Å |
| CryoDRGN | 3.81 Å | 4.02 Å | 3.66 Å | 3.27 Å |
| Ours | **2.98 Å** | **2.56 Å** | **2.62 Å** | **2.43 Å** |

reconstruction, followed by homogeneous refinement. For CryoDRGN, we employ the `train_nn` function, which is designed for homogeneous reconstruction without modeling heterogeneity. In both cases, we use their default parameters. In the following sections, we first evaluate the reconstruction efficiency, and then examnine the reconstructed protein densities from `GEM`, CryoSPARC and Cryo-DRGN with three mainstream evaluation protocols: gold-standard Fourier shell correlation (Harauz & van Heel (1986), Section 4.2), local resolution estimation (Adams et al. (2010), Section 4.3), and Fourier slice correlation (Huang et al. (2024), Section 4.4).

### 4.1 EFFICIENCY COMPARISON

We first evaluate the efficiency of `GEM` against deep learning-based cryo-EM reconstruction methods. Specifically, we compare with CryoDRGN (Fourier-based) and CryoNeRF (NeRF-based). All experiments are conducted on a single NVIDIA RTX 6000 GPU, and we report both training throughput (images per second) and peak GPU memory usage. As shown in Table 1, `GEM` consistently achieves the best efficiency, with substantially lower memory usage and faster speed than the baselines. CryoNeRF suffers from cubic memory and computation overhead inherent to NeRF, resulting in very slow training and frequent out-of-memory (OOM) failures. CryoDRGN, which leverages the Fourier slice theorem with quadratic complexity, is faster than CryoNeRF but still significantly outperformed by `GEM`. Moreover, `GEM` exhibits more stable memory usage, as it depends primarily on the number of 3D Gaussians used to represent the protein density.

### 4.2 GOLD-STANDARD FOURIER SHELL CORRELATION

Gold-standard Fourier shell correlation (GSFSC) (Van Heel & Schatz, 2005) is a commonly used evaluation to quantify the reconstruction quality of a cryo-EM reconstruction method on a dataset.

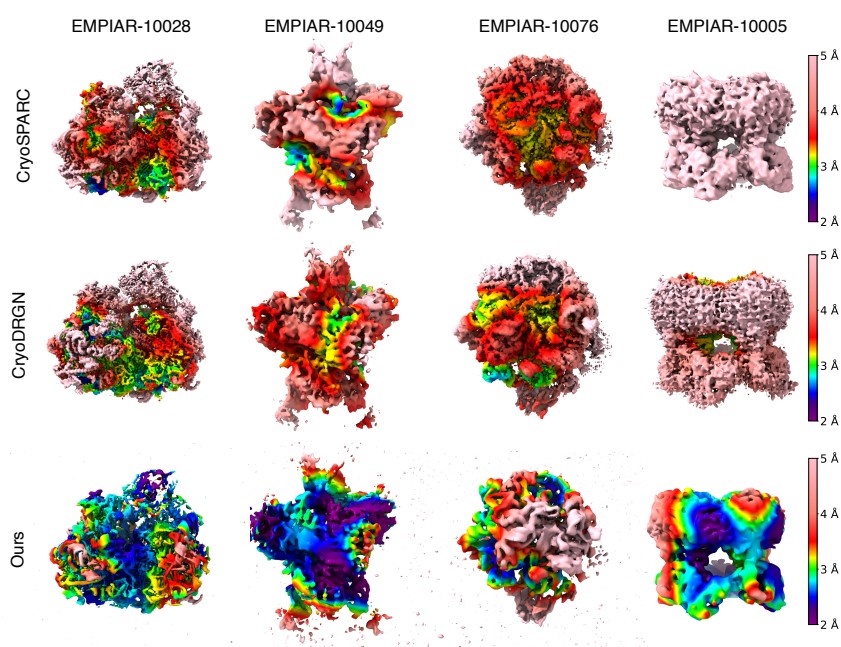

Figure 6: Local resolution maps of reconstructions from GEM, CryoSPARC, and CryoDRGN. Each map is colored according to the estimated local resolution, with the color scale ranging from 2 Å (blue, higher resolution) to 5 Å (red, lower resolution). **Better reconstructions are indicated by larger regions in blue-green.**

Table 3: Mean local resolution (in Å) of reconstructions on the four cryo-EM datasets. **Lower values correspond to higher resolution.**

|  | EMPIAR-10028 | EMPIAR-10049 | EMPIAR-10076 | EMPIAR-10005 |
|---|---|---|---|---|
| CryoSPARC | 3.55 Å | 3.57 Å | 3.16 Å | 5.54 Å |
| CryoDRGN | 3.39 Å | 3.22 Å | 3.22 Å | 3.38 Å |
| Ours | **2.87 Å** | **2.62 Å** | **2.99 Å** | **2.58 Å** |

Given a cryo-EM particle image dataset $\mathcal{I} = \{I_i\}_{i=1}^N$, to evaluate a method with GSFSC, the dataset $\mathcal{I}$ is first randomly splitted into two non-overlapping splits $\mathcal{I}_1$ and $\mathcal{I}_2$. Then the method to test is applied on the two splits to produce two reconstructions $X_1$ and $X_2$. The FSC value at spatial frequency shell $r_i$ in the Fourier space is defined as:

$$\text{FSC}(r_i) = \frac{\sum_{r \in r_i} F_1(r) \cdot F_2(r)^*}{\sqrt{\sum_{r \in r_i} |F_1(r)|^2 \cdot \sum_{r \in r_i} |F_2(r)|^2}}, \tag{9}$$

where $F_1(r)$ and $F_2(r)$ denote the Fourier coefficients of $X_1$ and $X_2$ at frequency $r$, and $^*$ indicates complex conjugation. The summations are computed over all voxels in the shell $r_i$. As in the GSFSC protocol the particle images are split into two halves $\mathcal{I}_1$ and $\mathcal{I}_2$ and reconstructed independently, it ensures that the correlation reflects reproducible structural information rather than overfitting.

Since the spatial frequency $r$ is inversely related to the 3D Euclidean space resolution, the GSFSC values can be plotted as a curve of FSC values versus resolutions (in Å). The resolution estimate is then determined by the point at which the GSFSC curve drops below a predefined threshold (commonly 0.143). This crossing point represents the highest spatial frequency (lowest resolution value) at which the reconstruction remains reliable, thereby providing a quantitative resolution assessment.

As shown in Figure 5 and Table 2, ❶ GEM consistently achieves higher resolutions (i.e., lower Å values at the intersection with the 0.143 FSC threshold) than the baselines across all datasets. It is

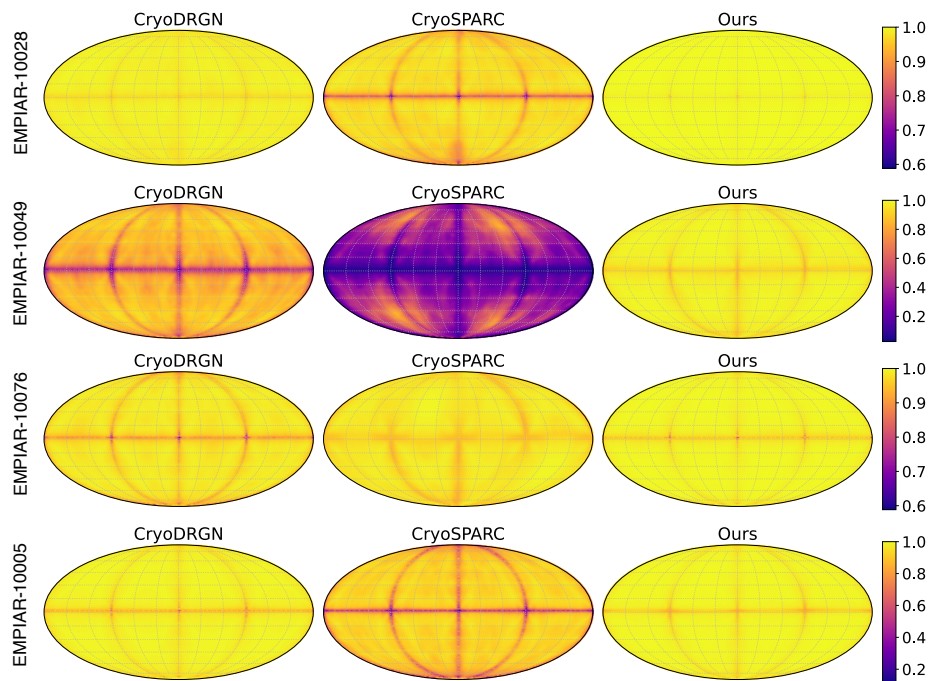

Figure 7: FSLC maps of reconstructions from GEM, CryoSPARC, and CryoDRGN across four datasets. Each map depicts the correlation values of slices sampled across different elevation and azimuth angles, with the color scale ranging from low correlation (purple) to high correlation (yellow). **More uniform yellow regions indicate higher and more isotropic directional resolution.**

Table 4: FSLC results on four cryo-EM datasets. Each row reports the mean FSLC value with its standard deviation across randomly sampled orientations. **Higher mean values indicate stronger directional consistency, and lower standard deviation reflects greater stability.**

|  | EMPIAR-10028 | EMPIAR-10049 | EMPIAR-10076 | EMPIAR-10005 |
| --- | --- | --- | --- | --- |
| CryoSPARC | $0.961 \pm 0.030$ | $0.348 \pm 0.160$ | $0.978 \pm 0.012$ | $0.886 \pm 0.084$ |
| CryoDRGN | $0.989 \pm 0.005$ | $0.814 \pm 0.109$ | $0.976 \pm 0.020$ | $0.974 \pm 0.025$ |
| Ours | $\mathbf{0.998 \pm 0.002}$ | $\mathbf{0.971 \pm 0.022}$ | $\mathbf{0.992 \pm 0.010}$ | $\mathbf{0.982 \pm 0.019}$ |

noteworthy that, according to the Nyquist-Shannon sampling theorem (Shannon, 2006), the ultimate attainable resolution is limited to twice the voxel size of the protein density map, corresponding to 2.46 Å and 2.62 Å for EMPIAR-10049 and EMPIAR-10028, respectively. ❷ On these two datasets, GEM achieves resolutions that approach this theoretical upper bound, highlighting its strong ability to recover fine structural details from cryo-EM images.

### 4.3 LOCAL RESOLUTION ESTIMATION

We then evaluate the reconstructions from all methods using the standard protocol of local resolution estimation. While the FSC provides a single global resolution value, the same principle can be localized to regions within the protein density, providing more detailed estimation of the quality of the reconstruction. Specifically, given two half maps reconstructed from independent data splits, a cubic subregion is extracted at the same location from each map. The FSC between the two cubes is then computed following Equation 9, and the local resolution is determined from the frequency at which the FSC curve falls below the $0.143$ threshold. This estimated resolution is assigned to the voxel at the cube center. Repeating this process across the entire map yields a resolution map that depicts the spatial variability of the reconstruction quality.

As shown in Figure 6 and Table 3, GEM consistently achieves superior local resolution across all four datasets. The EMPIAR-10005 dataset poses a significant challenge to existing approaches due to its

Table 5: Ablation study on different parameters of 3DGS. "No Rotation" means the rotation of all 3D Gaussians are fixed at 0. "Isotropic Scale" indicates the scale matrix $\mathbf{S}_i$ is identical across the three axis of the 3D Gaussians, resulting in a spherical shape of the 3D Gaussians.

|  | EMPIAR-10028 | EMPIAR-10049 | EMPIAR-10076 | EMPIAR-10005 |
|---|---|---|---|---|
| ❶ No Rotation | 3.71 Å | 4.23 Å | 5.99 Å | 4.15 Å |
| ❷ Isotropic Scaling | 4.02 Å | 5.53 Å | 12.80 Å | 4.21 Å |
| ❶ + ❷ | 10.67 Å | 6.81 Å | 13.54 Å | 7.95 Å |
| Original | **2.98 Å** | **2.56 Å** | **2.62 Å** | **2.43 Å** |

high symmetry and structural complexity, leading CryoSPARC and CryoDRGN to achieve average local resolutions of approximately 5 Å and 3 Å, respectively. In contrast, GEM attains an average local resolution of 2.28 Å, which approaches the Nyquist-Shannon limit of 2.43 Å, thereby demonstrating its strong capability to recover fine structural details.

### 4.4 FOURIER SLICE CORRELATION

We further validate the reconstructions using a recently proposed evaluation protocol, Fourier slice correlation (FSLC, Huang et al. (2024)). FSLC serves as an intermediate-level metric that quantifies directional resolution anisotropy by measuring the similarity between corresponding slices extracted from two reconstructed protein densities (Tan et al., 2017; Huang et al., 2024). In practice, slices are sampled at uniformly distributed elevation angles ($0° - 180°$) and azimuth angles ($0° - 360°$).

As shown in Figure 7 and Table 4, GEM consistently achieves higher FSLC values, indicating stronger agreement across a wide range of rotation angles. While all methods exhibit relatively lower correlation around the azimuthal regions of the elevation axis, GEM still outperforms the baselines, demonstrating superior robustness in challenging orientations. Moreover, our method attains the lowest standard deviation across the four datasets, further highlighting its stability and consistency with respect to directional variations.

### 4.5 DESIGN CHOICES OF 3DGS FOR CRYO-EM

While 3DGS was developed for reconstruction of natural scenes, it is unclear which set of parameters are essential for cryo-EM reconstruction. We therefore ablate per-Gaussian rotation and anisotropic scaling. Concretely, we evaluate: ($i$) No Rotation, fix $\mathbf{R}_i = \mathbf{I}$; ($ii$) Isotropic Scaling, enforce $\mathbf{S}_i = s_i\mathbf{I}$, where $s_i$ is the distance of the $i$-th 3D Gaussian to its nearest neighbor; ($iii$) No Rotation and Isotropic Scaling (both constraints).

As shown in Table 5 (lower Å is better), ❶ all ablations degrade resolution across datasets. ❷ Enforcing isotropic scaling is consistently more harmful than removing rotation (e.g., on EMPIAR-10076, 2.62 Å to 12.80 Å vs. 2.62 Å to 5.99 Å), and combining both constraints cause the most harm to the performance. These results indicate that both per-Gaussian rotation and anisotropic scaling are critical for accurately modeling cryo-EM densities, validating our choice to retain the full 3DGS parameterization in GEM.

## 5 CONCLUSION

In this work, we introduced GEM, a novel framework that leverages 3D Gaussian Splatting (3DGS) for cryo-EM reconstruction. By explicitly representing protein densities with 3D Gaussians, GEM avoids the information loss of Fourier-space methods and the cubic memory and computation overhead of NeRF-based real-space approaches, thereby achieving high resolution and efficiency. Through comprehensive experiments, GEM shows that it delivers up to $48\times$ faster training, $12\times$ lower memory consumption, and up to $38.8\%$ improvement in local resolution over prior state-of-the-art methods, often approaching the physical resolution limit of the microscope. Overall, GEM establishes 3DGS as a practical and scalable paradigm for cryo-EM, unifying efficiency with high-resolution accuracy and offering broad benefits to the structural biology community.

ETHICS STATEMENT

We strictly follow the ICLR Code of Ethics. This work relies solely on existing open-source cryo-EM datasets and does not involve any biological experiments. As such, it does not generate any harmful insights or biological outcomes.

REPRODUCIBILITY STATEMENT

Reproducibility is a core principle of this work. All datasets used are publicly available and can be accessed through their official repositories. The code, along with detailed instructions for its use, will be released as open source upon acceptance to ensure full reproducibility of our results.

THE USE OF LARGE LANGUAGE MODELS (LLMS)

To enhance clarity and readability, we employed OpenAI's GPT-5 and GPT-5-thinking models exclusively as language polishing tools. Their role was limited to proofreading, grammatical correction, and stylistic refinement—functions comparable to those of conventional grammar checkers and dictionaries. These tools did not contribute any new scientific content or ideas, and their usage is consistent with standard practices in manuscript preparation.

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

## 6 APPENDIX

### 6.1 DESCRIPTION OF BASELINE METHODS

**Fourier Slice Theorem for Reconstruction.** A common strategy in cryo-EM reconstruction (Zhong et al., 2021; Punjani et al., 2017) is to leverage the Fourier slice theorem (Bracewell, 1956), which states that the Fourier transform of a 2D projection of a 3D density $V_i$ corresponds to a central slice of its 3D Fourier transform perpendicular to the projection direction. Formally,

$$\mathcal{F}(\text{Proj}(V_i;\ \phi_i, t_i)) = \text{Slice}(\mathcal{F}(V_i);\ \phi_i, t_i), \tag{10}$$

where $\mathcal{F}(\cdot)$ denotes the Fourier transform and $\text{Slice}(\cdot; \phi_i, t_i)$ extracts the 2D slice orthogonal to the projection direction specified by $(\phi_i, t_i)$. This theorem implies that reconstructing the protein density reduces to learning to predict individual slices in Fourier space, thereby significantly lowering the memory footprint during training.

**Real-Space Cryo-EM Reconstruction.** The reconstruction formulation in Equation 3 closely parallels 3D reconstruction problems in computer vision, which has motivated recent work to perform cryo-EM reconstruction directly in real space using neural radiance fields (NeRFs) (Huang et al., 2024; Qu et al., 2025). Unlike Fourier-based methods, this approach avoids repeated Fourier transforms and can in principle achieve higher resolution. However, it requires predicting the full density $\hat{V}_i$ before supervision can be applied, since the contrast transfer function (CTF) $C_i$ must operate on the complete projection image due to the convolution operation.

Concretely, given a set of predefined coordinates $D$, each particle's pose is applied as

$$D_i = \text{Rotate}(D;\ \phi_i) + t_i, \tag{11}$$

and the resulting coordinates are used to query the NeRF, producing the predicted density volume $\hat{V}_i$. The model is then trained by projecting $\hat{V}_i$ under the same pose and computing the loss between the simulated projection $\hat{I}_i$ and the experimental image $I_i$:

$$\mathcal{L}_i = \text{MSE}\Big(I_i, \text{Proj}\Big(\hat{V}_i;\ \phi_i, t_i\Big)\Big). \tag{12}$$

Because the entire density $\hat{V}_i$ must be generated at each iteration, this method incurs cubic memory and computation cost, making training slow and resource-intensive despite its resolution advantage.

### 6.2 DERIVATION OF 3DGS PROJECTION

The $j$-th 3D Gaussian is defined as:

$$G_j(\mathbf{x}\,|\,\rho_j, \mathbf{p}_j, \boldsymbol{\Sigma}_j) = \rho_j \cdot \exp\left(-\frac{1}{2}(\mathbf{x} - \mathbf{p}_j)^\top \boldsymbol{\Sigma}_j^{-1}(\mathbf{x} - \mathbf{p}_j)\right). \tag{13}$$

Therefore for a query position $\mathbf{x}$ in the protein density, its value is defined as

$$\hat{V}_i(\mathbf{x}) = \sum_{j=1}^{M} G_j(\mathbf{x}\,|\,\rho_j, \mathbf{p}_j, \boldsymbol{\Sigma}_j). \tag{14}$$

To avoid the query of the entire protein density before the projection, we aim to explicitly derive the expression of the projected image $\hat{I}_i$. Since each pixel in the image $\hat{I}_i$ corresponds to an electron beam $\mathbf{r}$, we denote the projection of this specific pixel as $\hat{I}_i(\mathbf{r})$.

Following the definition, the projection of the pixel can be defined as:

$$\hat{I}_i(\mathbf{r}) = \int \hat{V}_i(\mathbf{x})\,dz = \int \sum_{j=1}^{M} G_j(\mathbf{x}\,|\,\rho_j, \mathbf{p}_j, \boldsymbol{\Sigma}_j)\,dz \tag{15}$$

$$= \sum_{j=1}^{M} \int G_j(\mathbf{x}\,|\,\rho_j, \mathbf{p}_j, \boldsymbol{\Sigma}_j)\,dz. \tag{16}$$

From the standard 3DGS for natural images, we have

$$\hat{I}_r(\mathbf{r}) \approx \sum_{j=1}^{M} \int G_j(\mathbf{x}|\rho_j, \mathbf{p}_j, \underbrace{\mathbf{J}_j \mathbf{W} \mathbf{\Sigma}_j \mathbf{W}^\top \mathbf{J}_j^\top}_{\tilde{\mathbf{\Sigma}}_i}) \, dz, \tag{17}$$

where $\tilde{\mathbf{\Sigma}}_i$ is the new Gaussian covariance matrix controlled by local approximation matrix $\mathbf{J}_j$ and viewing transformation matrix $\mathbf{W}$ that only corresponds to the rotation $\phi$. The projection of natural images follows the pinhole camera model, while in cryo-EM the projection rays are parallel, therefore the Jacobian matrix satifies $\mathbf{J}_j = \mathbf{I}$, which gives $\tilde{\mathbf{\Sigma}}_j = \mathbf{\Sigma}_j$. Thus following (Zwicker et al., 2002; 2001) we have

$$\hat{I}_i(\mathbf{r}) = \sum_{j=1}^{M} \rho_j (2\pi)^{\frac{3}{2}} \left| \tilde{\mathbf{\Sigma}}_j \right|^{\frac{1}{2}} \int \frac{1}{(2\pi)^{\frac{3}{2}} \left| \tilde{\mathbf{\Sigma}}_j \right|^{\frac{1}{2}}} \exp\left( -\frac{1}{2} \left( \mathbf{x} - \tilde{\mathbf{p}}_j \right)^\top \tilde{\mathbf{\Sigma}}_i^{-1} \left( \mathbf{x} - \mathbf{p}_j \right) \right) \, dz \tag{18}$$

$$= \sum_{j=1}^{M} \rho_j (2\pi)^{\frac{3}{2}} |\mathbf{\Sigma}_i|^{\frac{1}{2}} \int \frac{1}{(2\pi)^{\frac{3}{2}} |\mathbf{\Sigma}_i|^{\frac{1}{2}}} \exp\left( -\frac{1}{2} \left( \mathbf{x} - \mathbf{p}_i \right)^\top \mathbf{\Sigma}_i^{-1} \left( \mathbf{x} - \mathbf{p}_i \right) \right) \, dz \tag{19}$$

$$= \sum_{j=1}^{M} \rho_j \frac{1}{2\pi \left| \hat{\mathbf{\Sigma}}_j \right|^{1/2}} \exp\left( -\frac{1}{2} \left( \hat{\mathbf{x}} - \hat{\mathbf{p}}_j \right)^\top \hat{\mathbf{\Sigma}}_j^{-1} \left( \hat{\mathbf{x}} - \hat{\mathbf{p}}_j \right) \right). \tag{20}$$

## 6.3 DESCRIPTION OF DATASETS

The characteristics of datasets used in this paper are summarized as follows:

- **EMPIAR-10005** (TRPV1): A well-characterized protein in the vertebrate TRP family, and is frequently used to investigate fundamental TRP channel functions and structures.
- **EMPIAR-10028** (*Pf* 80S ribosome): Contains a complex composition of the antibiotic and *Pf* ribosome and is commonly used for evaluation of the reconstruction resolution.
- **EMPIAR-10049** (Synaptic RAG1–RAG2 complex): Exhibits substantial compositional and conformational heterogeneity and is therefore challenging for high-resolution reconstruction.
- **EMPIAR-10076** (L17-depleted 50S ribosomal intermediates): Comprises complex intermediate assembly states that can cause reconstruction quality degradation.

