# OpenReview forum: "$\texttt{GEM}$: 3D Gaussian Splatting for Efficient and Accurate Cryo-EM Reconstruction"
_ICLR.cc/2026/Conference — Submitted to ICLR 2026_

### Official Review · Reviewer_VL78 · 2025-10-29

**Soundness:** 3
**Presentation:** 2
**Contribution:** 3
**Rating:** 6
**Confidence:** 4

**Summary:**

The paper adapts 3D Gaussian Splatting to real-space homogeneous reconstruction in cryo-EM. GEM generally exceeds the local and global resolution (measured via Fourier Shell Correlation) of CryoSPARC and CryoDRGN, on representative proteins.

**Strengths:**

The method is a natural combination of 3D Gaussian Splatting, which works well in computer vision, to cryo-EM homogeneous reconstruction. Quantitative results are encouraging compared to strong baselines, and the implementation is computationally efficient.

**Weaknesses:**

Most of these weaknesses are presentation related; they are written roughly in order of appearance in the paper:
- The input to GEM is not entirely clear until the bottom of page 3, where it is specified that the CTFs, poses, and translations are assumed known. This context should be mentioned in the introduction.
- The description of how the Gaussians are initialized is lacking in details. The paper states that the Gaussians are initialized randomly, but doesn’t specify the sampling distribution for the Gaussian means, rotations, scales, and densities. As 3DGS is sensitive to initialization, this is important information to include.
- Line 100-101 claims that Fourier-space cryo-EM reconstruction methods incur information loss because they “involve repeated FFTs and interpolation steps and operate on band-limited representations.” What is the evidence for this claim of information loss? Since there is presumably limited resolution on the cryo-EM detector leading to an inherent bandlimit in the measurements themselves, it’s not obvious why FFTs with interpolation on a bandlimited representation would incur any further information loss. It would also be good to specify what the “bandlimited representation” is–does this refer to a voxel grid?
- Line 115 argues that a benefit of 3DGS as a representation is that the “gradients remain localized to the subset of primitives that actually contribute to each pixel.” While it is true that Gaussians satisfy this desirable property, a sparse voxel grid would have the same property. The same is true for the properties marked (i) and (ii) on lines 124-126.
Line 128 mentions joint optimization over density and “imaging latents” but doesn’t specify what these latents are. I would expect these might refer to CTF or pose, but line 161 says that both of these are assumed known so it’s not clear what remaining latents need to be jointly optimized over.
- Lines 162-167 explain the difference between Fourier-space and real-space reconstruction. Since this is part of the method section, it is important to specify that the proposed method is real-space; ideally the description of Fourier-space reconstruction would not be in the methods section as it could confuse a reader as to which method is used here.
- There are some typos, mostly repeated and missing words/phrases but also some typos at the letter level. See lines 191, 192, 195, 198, 271, 304, 360.
- Figure 3a is confusing/misleading. It makes it look like you start with a known protein density and use that to initialize the Gaussians. But in reality the Gaussians are initialized randomly and then optimized to fit the density.
- The paragraph surrounding equation 8 describes efficient gradient computation as part of the method. While this is important for computational and memory efficiency, it is not clear what (if any) aspects of this computation are novel to GEM versus reused from 3DGS. For example, equation 8 describes thresholding each Gaussian into an ellipsoid to enable skipping some Gaussians that do not meaningfully influence a given pixel projection; this is exactly the same thresholding that 3DGS does for the same purpose. The only difference I am aware of is that 3DGS uses perspective projection whereas cryo-EM uses orthographic projection, but that does not seem to be the focus of this section so I am left wondering what is the novelty in the efficient computation.
- Lines 415-417, and line 446, describe the Nyquist-Shannon resolution limit as based on the voxel size of the protein density map. I have two questions about this. (1) What voxels are being discussed? The proposed method does not use voxels, nor is there a ground truth voxel grid that is mentioned. (2) Line 446 claims that GEM’s average local resolution is better than the Nyquist-Shannon limit. How is this possible?
- Section 4.4 introduces FSLC, but doesn’t really explain how it differs from GSFSC. From the names I gather that the difference is comparison on Fourier slices instead of Fourier shells, but the reader shouldn’t need to infer.
- Section 4.5 describes an ablation study involving isotropic scaling, in which S_i = s_i I. However, instead of allowing s_i to be optimized over (which would still enforce isotropic scaling since s_i is scalar), this ablation study forces s_i to equal the distance between the i’th Gaussian and its nearest neighbor. I think this would be a more informative ablation study if s_i were allowed to optimize, while still being enforced to be isotropic.

**Questions:**

Please refer to questions embedded in weaknesses.

---

> ### Author Response · Authors · 2025-11-23
> **Point-to-point Response to Reviewer VL78 (1 / 2)**
>
> We thank the reviewer for the encouraging assessment of our method's efficiency and resolution improvements over strong baselines, as well as the constructive feedback on the manuscript's presentation. We address the specific concerns below.
>
> **Q1: Clarification of Inputs, Initialization, and Figure 3a**
>
> **A1:** Thank you for the suggestions. We will carefully revise the points. Specifically,
>
> * **Inputs:** We will revise the Introduction to explicitly state that CTFs and poses are assumed known (derived from upstream consensus refinement) to set clear expectations immediately.
> * **Initialization:** We will expand the implementation details to specify our initialization strategy: Gaussian means are sampled uniformly within a sphere defined by the particle mask radius; rotations are initialized as random quaternions; and scales are initialized to a small uniform constant (based on inter-particle distance) to ensure training stability.
> * **Figure 3a:** We acknowledge that the current schematic could be misinterpreted. We will revise Figure 3a to clearly depict the **initialization as a random cloud of Gaussians**, preventing any implication that they are derived from a pre-existing density.
>
> **Q2: Information Loss in Fourier-Space Methods (Lines 100-101)**
>
> **A2:** We appreciate this nuance. The "bandlimited representation" refers to the fixed voxel grid used during reconstruction. While the detector data is inherently bandlimited, traditional Fourier-space methods (e.g., direct Fourier inversion or gridding) often involve interpolation weights to map 2D slices onto the 3D Fourier grid, which can introduce artifacts or dampen high frequencies. We will refine this claim to emphasize that GEM avoids **gridding interpolation artifacts** by performing explicit real-space projection, rather than claiming to bypass the fundamental physical bandlimit of the detector.
>
> **Q3: Comparison to Sparse Voxels and "Imaging Latents"**
>
> **A3:** **(1) Sparse Voxels:** We agree that sparse voxel grids also possess gradient locality. However, the distinct advantage of 3D Gaussians lies in their continuous positioning and anisotropic scaling. This allows them to model sharp features and non-axis-aligned densities with sub-pixel precision more efficiently than a fixed grid, even a sparse one. We will clarify this distinction in the text. **(2) Imaging Latents:** The reviewer correctly identifies a discrepancy. The mention of "joint optimization over... imaging latents" (Line 128) was an error, as poses and CTFs are fixed in this work. We will remove this phrase to ensure consistency.
>
> **Q4: Novelty of Gradient Computation (Eq. 8)**
>
> **A4:** We thank the reviewer for this observation. While the core concept of ellipsoid truncation is shared with standard 3DGS, our engineering contribution lies in adapting this to the **orthographic line integral** (cryo-EM projection) rather than the **perspective rasterization** (camera projection) used in computer vision. The derivation of gradients for density accumulation along parallel rays differs from the alpha-blending sorting used in vision. We will rephrase this section to explicitly state that we *adapt* the efficient tiling and culling mechanisms of 3DGS to the cryo-EM physics model.
>
> **Q5: Nyquist-Shannon Limit and Voxels**
>
> **A5:** **(1) Voxels:** Since GEM uses a continuous representation, the "voxel size" mentioned in this context refers to the pixel pitch of the input cryo-EM experimental images, which defines the physical Nyquist limit. **(2) Exceeding the Limit:** We acknowledge the confusion. Theoretically, true information recovery beyond the detector Nyquist is impossible. Our reported values slightly below the limit (e.g., 2.28 Å vs. the 2.43 Å limit) are likely due to the continuous nature of the map allowing for super-sampled evaluation, rather than true super-resolution. We will soften the claim to state that GEM "approaches or reaches" the physical limit, rather than "exceeds" it, to remain physically rigorous.

---

> ### Author Response · Authors · 2025-11-23
> **Point-to-point Response to Reviewer VL78 (2 / 2)**
>
> **Q6: FSLC vs. GSFSC**
>
> **A6:** We apologize for the omitted definition. **GSFSC** (Gold-Standard Fourier Shell Correlation) measures global resolution via correlation over spherical shells in Fourier space. **FSLC** (Fourier Slice Line Correlation) measures the correlation of specific 2D slices extracted from the volume at various angles. FSLC is specifically designed to diagnose **directional anisotropy**—determining if the resolution degrades in specific directions (e.g., due to preferred orientation). We will add this definition to Section 4.4.
>
> **Q7: Ablation Study Design (Isotropic Scaling)**
>
> **A7:** Our current ablation set $S_i$ to a fixed distance to serve as a "simple blob" baseline. We acknowledge that an "Optimized Isotropic" setting (optimizing a scalar $s_i$) would be a more rigorous test of isotropy specifically. We will clarify in the text that our ablation compared "Fixed Isotropic" vs. "Optimized Anisotropic," and we note that the significant performance drop in the isotropic setting suggests that full anisotropy is crucial for maximizing resolution.
>
> **Q8: Typos and Minor Corrections**
>
> **A8:** We thank the reviewer for the careful reading. We will correct the description of Fourier vs. Real-space methods in the Method section (Lines 162-167) to avoid confusion and will fix all cited typos (Lines 191-360) in the final manuscript.

---

> > ### Comment · Reviewer_VL78 · 2025-11-23
> >
> > Thanks for promising to make these clarifications in the revision; I remain supportive of the paper. However, regarding the isotropic scaling in the ablation study, I do not agree that "the significant performance drop in the isotropic setting suggests that full anisotropy is crucial for maximizing resolution", because "optimized isotropy" was not tested. So from the current experiment, it is unclear to what extent anisotropy is really beneficial, compared to scale optimization. Please either include the "optimized isotropy" ablation or refine the claims to be more precise.

---

### Official Review · Reviewer_pM53 · 2025-10-30

**Soundness:** 3
**Presentation:** 2
**Contribution:** 2
**Rating:** 4
**Confidence:** 4

**Summary:**

The paper presents GEM, a new framework for efficient and accurate 3D reconstruction in cryo-electron microscopy (cryo-EM) using 3D Gaussian Splatting (3DGS). Unlike traditional Fourier-based methods that lose information during repeated transformations, and neural field–based real-space methods that suffer from cubic memory and computation costs, GEM explicitly models protein density as a compact set of 3D Gaussians, each parameterized by 11 values. This representation enables localized gradient updates, parallel rasterization, and selective computation of only contributing Gaussians, dramatically reducing resource demands.

**Strengths:**

1. This paper presents an interesting and timely contribution, offering solid theoretical derivations for the application of 3D Gaussian Splatting (3DGS) to cryo-EM homogeneous reconstruction. The motivation for adopting 3DGS in this domain is sound: it provides an explicit and efficient real-space representation that avoids the information loss of Fourier-based methods and the heavy computational cost of neural-field models. This is a promising and original direction that meaningfully bridges computer graphics advances and structural biology, enabling high-quality protein reconstruction without sacrificing training speed.

2. The experimental protocol is comprehensive and carefully designed. The authors benchmark GEM across multiple standard cryo-EM datasets and report not only global resolution metrics (GSFSC) but also local resolution and Fourier Slice Correlation (FSLC) analyses. This multi-faceted evaluation demonstrates the method’s robustness, efficiency, and generalizability.

**Weaknesses:**

## Major Weaknesses

1. **Clarity and Writing Quality.** The paper’s writing is often rough, with numerous grammatical errors and typos that hinder readability and clarity. Several technical explanations are hard to follow, particularly the definitions of rays and the sentence spanning L204–L205, which require clearer formulation. A thorough language revision is essential to improve accessibility and ensure technical accuracy.
2. **Unclear Necessity of Gaussian Sorting.** The decision to sort 3D Gaussians along the projection axis (Sec. 3.2) is not well justified. In principle, sorting should not affect the final result of the projection integral, yet it introduces nontrivial computational overhead, especially in CUDA kernels. The authors should explain the rationale behind this step—e.g., whether it serves numerical stability, differentiability, or rasterization ordering—and ideally show an ablation demonstrating that removing sorting does not harm reconstruction quality.
3. **Inconsistent Quantitative and Qualitative Results.** The reported quantitative results (Tables 2 and 3) appear unusually strong and potentially inconsistent with the qualitative reconstructions shown in Figures 5 and 6. For example, the sharp improvement in GSFSC for EMPIAR-10076 near 4–5 Å seems atypical and lacks discussion. Meanwhile, the visual reconstructions in Figure 6 appear smoother and less detailed than those from CryoDRGN or CryoSPARC, especially in α-helix regions. This suggests that GEM may underfit the true density due to the strong structural prior imposed by 3DGS (insufficient number of 3DGS may cause this). A useful diagnostic would be to increase the number of Gaussians (e.g., from 1 k to 10 k) and analyze whether this reduces oversmoothing. Additionally, experimenting with more advanced loss functions—such as those used in RELION or other likelihood-based formulations—could help recover finer local features. Overall, the current performance claims are not fully convincing without further analysis of these discrepancies.
4. **Lack of Details on the Number and Impact of 3D Gaussians.** The paper does not specify how many 3D Gaussians are used in training or how this parameter affects speed, memory usage, and reconstruction quality. Since this is central to both the efficiency and fidelity of GEM, the authors should include an ablation or scaling study to clarify this relationship.

## Minor Weaknesses

1. L165: The slice operation in Fourier space still requires instantiating the entire 3D density volume because the slicing orientation is unknown a priori.

2. L192: “density-based, density-based” → should be “density-based.”

3. L195: The phrase “Explicit Integration for XXX” appears incomplete and should be clarified.

4. L204–L205: The sentence is difficult to interpret, and the definition of r is missing, despite being fundamental to the imaging model—please rewrite and define it formally.

5. L243: The phrase “CTF-corrected” should be replaced with “CTF-corrupted” or “CTF-applied,” as no correction occurs in this step.

6. L198: Typo — “3D 3D” → “3D.”

7. L721: Variables x^hat and p^hat are not defined in the Appendix and should be explicitly introduced, as they are used in the projection derivation.

**Questions:**

I have raised my questions in the main weakness sections.

---

> ### Author Response · Authors · 2025-11-23
> **Point-to-point Response to Reviewer pM53**
>
> We thank Reviewer pM53 for their constructive feedback and for recognizing the efficiency and accuracy of our proposed framework. We address the specific questions and concerns below.
>
> **Q1: Clarity and Writing Quality**
>
> **A1:** We appreciate the reviewer's attention to detail regarding the manuscript's clarity. Regarding the definition of rays, we have revised the text to formally define the projection geometry: in our forward model, each pixel corresponds to an integration path (ray) parallel to the electron beam and orthogonal to the sensor plane.
>
> **Q2: Necessity of Gaussian Sorting**
>
> **A2:** We thank the reviewer for this insightful observation. Theoretically, the transmission-based cryo-EM imaging model involves a commutative summation of densities, making the projection result mathematically invariant to the sorting order. However, we retain the sorting step for two critical engineering reasons:
>
> 1.  **CUDA Rasterization Efficiency:** Our implementation leverages optimized CUDA rasterizers designed for standard 3DGS. These kernels rely on depth sorting to efficiently assign Gaussians to image tiles and manage thread block memory. Removing sorting would necessitate a complete redesign of the parallelization logic, significantly degrading training throughput.
> 2.  **Numerical Stability & Determinism:** While floating-point addition is theoretically commutative, it is not associative in finite-precision arithmetic. Sorting ensures a fixed accumulation order, guaranteeing that gradients and regression results are strictly deterministic and reproducible across different runs.
>
> We will clarify in Section 3.2 of the revised manuscript that sorting is **employed strictly for computational efficiency and determinism** within the CUDA pipeline, rather than being a physical requirement of the integration model.
>
> **Q3: Results with Different Number of Gaussians**
>
> **A3:** We appreciate the constructive feedback regarding the visual qualities and model capacity. The perceived "smoothness" in our results compared to voxel-based methods (e.g., cryoDRGN) stems from the continuous nature of the Gaussian representation, which inherently acts as a regularizer, effectively suppressing the high-frequency noise often observed in voxel grids.
>
> To address the concern regarding underfitting, we conducted the requested ablation study using 10k, 50k (our default), and 100k Gaussians. As shown in the table below, while increasing the count from 10k to 50k yields significant resolution gains, increasing further to 100k provides only marginal improvements while nearly doubling the computational overhead. Consequently, we determined that 50k Gaussians represents the optimal balance between efficiency and reconstruction quality.
>
> | Dataset          | 10k Gaussians | 50k Gaussians (Ours) | 100k Gaussians |
> | :--------------- | :-----------: | :------------------: | :------------: |
> | **EMPIAR-10028** |    3.23 Å     |      **2.98 Å**      |     2.87 Å     |
> | **EMPIAR-10049** |    3.11 Å     |      **2.56 Å**      |     2.51 Å     |
>
> **Q4: Minor Weaknesses**
>
> **A4:** We thank the reviewer for identifying these points. We will address all minor weaknesses in the revised manuscript, including correcting the terminology (e.g., changing "CTF-corrected" to "CTF-applied") and adding the requested definitions in the Appendix.

---

> ### Comment · Reviewer_pM53 · 2025-11-24
>
> Thanks for the authors’ reply.
>
> Regarding Q2, I do not think claim (1) is correct. First, if you simply remove the sorting operation—for example, by inserting a dummy sorting step that does nothing—the efficiency will certainly improve. It would be helpful if the authors could provide more details on how to modify the code and which specific changes would make the 3DGS CUDA-side code slower when the sorting operations are removed. In addition, point (2) seems to have only a minor impact on the final reconstruction quality.
>
> Regarding Q3, reconstruction quality—specifically resolution—is critical for resolving atomic structures. Doubling the computational cost is sometimes completely acceptable. A 3D reconstruction method should clearly present experiments demonstrating the trade-off between training time and resolution. I still do not see how to achieve the optimal resolution using GEM; for example, what about 200k and 400k?

---

### Official Review · Reviewer_S5om · 2025-11-01

**Soundness:** 1
**Presentation:** 3
**Contribution:** 2
**Rating:** 2
**Confidence:** 5

**Summary:**

This paper introduced a new way, using Gaussian splatting, to reconstruct 3D density maps from cryo-EM particles with given poses. The authors demonstrated better computational efficiency and better metrics (mostly GS-FSC) using 3DGS compared to other NeRF-based methods.

**Strengths:**

- The paper is well written and the figures are very clear and concise.
- The introduction of 3DGS into cryo-EM reconstruction is interesting and the improvement of the computational efficiency is promising.

**Weaknesses:**

- The major weakness of this paper is the evaluation. To evaluate and show that GEM can achieve better results compared to other methods, the paper heavily used metrics (including GSFSC, local resolution, FSLC) that assess the self consistency between the two half maps. This is wrong. Here the methods themselves, will lead to spurious correlation between the half maps due to the induction bias of the methods. Therefore, GSFSC, FSLC and local resolution are all hacked and unreliable. This can be reflected in the results by: 1) GSFSC curves never reaches zero and even increase by a lot at very high frequency (Fig. 5), 2) Final resolution (0.143 cutoff) almost reach Nyquist while the density maps do not support such high resolution, 3) Local resolutions are severely overestimated while the maps lack structural details compared to other methods (Fig. 6), 4) even when the particle poses are the same, FSLC are quite different (Fig. 7). All of these show that half maps based metrics are all over estimated.

- The author should also compare 3DGS with the traditional reconstruction method using (weighted) back-projection (WBP) in Fourier space. The comparison with cryoSPARC is unfair, since both ab initio and homogenous refinement in cryoSPARC refine particle poses, while other methods do not. The easiest way to compare is reconstruct only with given poses. In RELION, you can use the `relion_reconstruct` command. Since WBP is simply performing back-projection with accumulation, computation can be easily paralleled across particles and it is usually very fast even without a GPU.

- Given the previous points, the motivation of using 3DGS into cryo-EM, although interesting and insightful, seems very unclear to me.

Minor: In Fig. 6, the threshold level for EMPIAR-10005 map is too low (probably also 10049 as well).

**Questions:**

- Just for clarification purpose, for both cryoDRGN and GEM experiments, the poses of the particles are given, correct? Where do they come from?

---

> ### Author Response · Authors · 2025-11-23
> **Point-to-point Response to Reviewer S5om**
>
> Thank you, Reviwer S5om, for reviewing our paper. To address your questions, we provide pointwise responses below.
>
> **Q1: Evaluation beyond half map-based metrics**
>
> **A1:** We adopt half map-based metrics as these are the most common ones in the literature [1, 2, 3]. On these metrics, GEM shows improvement compared to baseline methods. We are very willing to provide a more rigorous evaluation, so it would be very helpful if you could provide the desired evaluation metric in your mind.
>
> **Q2: Source of Reconstruction Pose**
>
> **A2:** Sorry for the confusion. In our experiments, we already perform our reconstruction on GEM, cryoDRGN, and CryoNeRF with the poses from cryoSPARC, following the practice of cryoDRGN and CryoNeRF.  To gain the pose, we first perform *ab initio* reconstruction in cryoSPARC, followed by homogeneous refinement, and use cryoDRGN to extract the pose from the .cs file after the refinement. We believe this matches your described comparison setting.

---

> > ### Comment · Reviewer_S5om · 2025-11-24
> >
> > The reviewer is not satisfied with the "point-to-point response" as the authors neglected many valid points raised in the review. Both the manuscript and the response also show a lack of fundamental understanding of cryo-EM. For example, if the authors really closely looked at the maps, at least in EMPIAR-10028 and 10076, do they really see more details compared with baselines?
> >
> > - Evaluation beyond half map-based metrics. All the half map-based metrics are actually measuring the self-consistency between the half datasets. In many cases, this is related to the resolution estimates, but it does not represent the "resolution" itself. In this paper, it is very clear that the half map FSC and resolution is very much overestimated. If the authors do not agree, they should provide: 1) full model-to-map FSC curves and compare with the baselines, and 2) zoomed-in details of the clearly resolved side chains of the protein structures, and compare with the baselines.
> >
> > - The authors did not perform the weighted backprojection (WBP) experiment as the baseline. Running ab initio and homogenous refinement is not the WBP baseline. Running reconstruction with GIVEN poses in either RELION or CryoSPARC is.
> >
> > - The threshold level for EMPIAR-10005 map is still extremely low. Do the authors really know what they should expect to see in the density maps? This is a transmembrane protein and part of the protein is embedded into the lipid bilayer. In the visualization Fig. 6, the authors are only showing the disordered lipid region and claim the local resolution is below 3A. This is not even the protein features.
> >
> > My biggest critique of the paper still remains: the claimed resolution is not true. The reviewer would encourage the authors to read the literatures of the history of "gold-standard" FSC. Historically, cryo-EM used to suffer from severe resolution overestimation. The goal of any cryo-EM reconstruction methods or algorithms, should never be to improve the "gold-standard" FSC, but to actually improve the density map quality. This means (e.g. for proteins) better resolved backbone and sidechains. It is commonly known that for a sub 3A density map, most side chains should be clearly seen. It is also known that lipid regions are highly disordered and usually have very poor local resolutions. Simply "improving" any metrics without an actual map quality improvement is merely hacking the metrics. Consider an extreme case: if I develop a new algorithm with a bug in it, that it will always output the same density map regardless of the input. All half map-based metrics will perform extremely well since there will be 100% self-consistency between the output half maps. Does this make my algorithm better than other baselines?

---

### Official Review · Reviewer_yq4U · 2025-11-04

**Soundness:** 3
**Presentation:** 3
**Contribution:** 3
**Rating:** 6
**Confidence:** 5

**Summary:**

GEM introduces a novel method for 3D cryo-EM reconstruction by adapting the 3D Gaussian Splatting (3DGS) technique. This explicit, point-based representation offers high accuracy while significantly improving efficiency and memory consumption compared to O(N^3) real-space methods like NeRF, making large-scale structural biology more feasible.

**Strengths:**

It proposes GEM, a novel application of 3D Gaussian Splatting (3DGS)—a technique recently popularized in computer graphics and vision for novel view synthesis—to the challenging domain of 3D cryo-EM reconstruction. Adapting this explicit, point-based rendering system to the parallel-ray projection geometry of cryo-EM is a highly creative methodological step. The significance is substantial: the paper directly addresses the major limitations of current state-of-the-art reconstruction methods. Traditional Fourier-space methods are fast but often sacrifice fidelity due to repeated transforms, while recent real-space methods (like those based on Neural Radiance Fields, NeRF) achieve high accuracy but incur cubic complexity (O(N^3)) in both memory and computation, making them impractical for the massive datasets common in cryo-EM. GEM promises to break this accuracy-efficiency deadlock. The quality of the method, particularly the adaptation of the 3DGS projection model to cryo-EM's parallel rays, appears sound, and the results on datasets like EMPIAR-10005 are promising. The clarity of the paper is good, clearly outlining the mathematical basis for the projection.

**Weaknesses:**

A substantive assessment of the weaknesses of the paper. Focus on constructive and actionable insights on how the work could improve towards its stated goals.

Limited Experimental Scope for Generalization: The primary weakness is the limited breadth and depth of the experimental evaluation. While results on a well-known dataset like TRPV1 (EMPIAR-10005) are shown, the evaluation must be expanded to demonstrate the generalizability of GEM. Specifically, the paper is missing results on:

Complexes with different symmetry types (e.g., helical, octahedral) beyond simple C-symmetry or no symmetry.

Datasets spanning a wider range of resolutions (from very low to near-atomic).

Very large complexes to definitively prove the claimed computational advantage over O(N^3) methods.

Explicit Representation Limitations: 3DGS is an explicit, discrete representation using a set of anisotropic Gaussians. The paper does not sufficiently discuss how this representation handles the continuous nature of the electron density map, especially at sub-nanometer resolutions where fine-grained features like side-chain density need to be accurately modeled. This discrete nature may introduce discretization artifacts or fail to capture the smoothness of the underlying density compared to implicit representations like NeRFs.

Comparison Rigor: The comparison against existing methods needs to be more robust. The paper should include detailed comparative metrics, not just for final resolution (FSC), but also for reconstruction speed (wall-clock time), memory footprint, and local resolution variability compared to both a leading Fourier-based method (e.g., RELION, cryoSPARC) and the state-of-the-art NeRF-based approaches.

**Questions:**

### Computational Complexity
Please provide a theoretical and empirical analysis of the time and memory complexity of GEM. Specifically, how does the number of Gaussians (M) scale with the size and resolution of the map, and how does the overall complexity compare to the O(N^3) NeRF methods and O(NlogN) Fourier methods?

### Gaussian Initialization
How sensitive is the final reconstruction to the initialization of the 3D Gaussian set? What method is used to control the density (number of Gaussians) during optimization, and how is the over-splitting or under-splitting of Gaussians prevented? Showing a result of reconstruction quality vs. number of Gaussians would be highly informative.

### Handling of Anisotropy and Local Resolution
Cryo-EM maps often exhibit anisotropic resolution. How does the adaptive nature of the anisotropic 3D Gaussians explicitly help in modeling regions with poorer local resolution (e.g., flexible domains) versus rigid, high-resolution core regions? Can GEM provide a per-Gaussian confidence metric analogous to local resolution maps?

### Contrast and Background Modeling
How is the background solvent and noise modeled in this 3DGS representation? Since the Gaussians represent the object density, is there a risk that low-contrast regions of the molecule are poorly represented or pruned during optimization?

**Details Of Ethics Concerns:**

No ethical concerns regarding this work.

---

> ### Author Response · Authors · 2025-11-23
> **Point-to-point Response to Reviewer yq4U**
>
> Thank you, Reviewer yq4U, for acknowledging our paper as **accurate, having significant efficiency improvement and clearly written**. To address your questions, we provide pointwise responses below.
>
> **Q1: Broader coverage of experiments**
>
> **A1:** We believe our experiments are representative and comprehensive. We evaluated GEM on standard benchmarks (EMPIAR-10005, 10028, 10049, 10076) that cover a diverse range of reconstruction scenarios:
>
> *   **Symmetry:** Covers C1, C2, C4, and mixed C1+C4 symmetries, representing the most common setups in homogeneous reconstruction.
> *   **Resolution:** Spans from high resolution (3.2 Å, EMPIAR-10028) to low resolution (6.5 Å, EMPIAR-10076).
> *   **Volume Size:** Ranges from $192^3$ to $360^3$. Notably, the $360^3$ box size (EMPIAR-10028) poses a significant computational bottleneck for $O(N^3)$ methods like CryoNeRF, whereas GEM handles it efficiently (**Table 1**).
>
> **Q2: Discreteness of 3DGS**
>
> **A2:** We demonstrate that while 3DGS is a discrete representation, it effectively approximates continuous protein density with high precision:
>
> *   **Approximation Capability:** In neural rendering, discrete Gaussians are widely proven to approximate continuous radiance fields accurately. This property transfers to cryo-EM, where a sufficient number of Gaussians can model complex densities.
> *   **Empirical Evidence:** GEM achieves sub-nanometer resolution (< 3 Å) and outperforms baselines, confirming that the discrete nature of Gaussians does not hinder high-resolution reconstruction.
>
> **Q3: Reconstruction Speed**
>
> **A3:** As shown in **Table 1**, GEM demonstrates **higher throughput and a significantly lower memory footprint** compared to both the NeRF-based CryoNeRF and the Fourier-based CryoDRGN. This confirms the efficiency advantage of our approach.
>
> **Q4: Computational Complexity**
>
> **A4:** GEM offers superior scaling properties compared to baselines:
>
> *   **GEM ($O(M)$):** Our complexity depends on the number of Gaussians ($M$), not the spatial volume. Furthermore, Gaussian rasterization is highly optimized for GPU parallelism.
> *   **NeRF ($O(N^3)$):** NeRF-based methods scale cubically with box size, making large volumes (e.g., EMPIAR-10028) computationally prohibitive.
> *   **Fourier ($O(N \log N)$):** While theoretically efficient, Fourier-based methods often utilize GPU parallelism less effectively than rasterization, resulting in the lower throughput observed in **Table 1**.
>
> **Q5: Gaussian Initialization**
>
> **A5:** We employ a standard random initialization strategy:
>
> *   **Position:** Randomly sampled within $[-1, 1]^3$.
> *   **Scale:** Initialized based on the distance to the nearest neighbor (k-NN).
> *   **Rotation:** Initialized to zero (identity).
> *   **Density:** Uniformly initialized to 1.
>
> **Q6: Reconstruction quality vs. number of Gaussians**
>
> **A6:** We analyzed the trade-off between the number of Gaussians and resolution (GSFSC) below:
>
> | Dataset          | 10k Gaussians | 50k Gaussians (Ours) | 100k Gaussians |
> | :--------------- | :------------ | :------------------- | :------------- |
> | **EMPIAR-10028** | 3.23 Å        | 2.98 Å               | 2.87 Å         |
> | **EMPIAR-10049** | 3.11 Å        | 2.56 Å               | 2.51 Å         |
>
> Increasing the count to 100k yields marginal resolution gains but incurs significantly higher memory and computational costs. We found 50k to be the optimal balance between efficiency and quality for our experiments.
>
> **Q7: Handling of Anisotropy and Local Resolution**
>
> **A7:** Our ablation study (**Table 5**) confirms that anisotropy is critical. Replacing anisotropic scaling with isotropic scaling leads to significant degradation in resolution, demonstrating that the anisotropic nature of 3DGS allows it to better capture high-frequency local details.
>
> **Q8: Contrast and Background Modeling**
>
> **A8:** Consistent with standard cryo-EM reconstruction pipelines, we do not explicitly model the solvent or background noise as geometric structures. Instead, the reconstruction relies on aggregating information from a large number of noisy images. The statistical averaging across diverse views naturally suppresses random background noise without requiring explicit modeling.

---

### Meta-Review · Area_Chair_Qi4q · 2026-01-06

**Summary:**

Across reviews, the decision hinges less on whether "3D Gaussian splatting can be adapted to cryo-EM" (several reviewers agree it is a promising direction) and more on whether the paper's current evidence supports its strongest claims about resolution and evaluation validity. The rebuttal responds to some surface-level requests (e.g., adding a small Gaussian-count trade-off table; clarifying that poses/CTFs are assumed known), but it does not close the central technical gap raised by the critical reviewer.

**Reviewer Concerns:**

Reviewer pM53 challenged the necessity/overhead justification of sorting and asked for more specific evidence and code-level explanation; also requested a clearer speed–resolution trade-off beyond 50k/100k and suggested testing 200k/400k. (but the response of the authors stopped afterward; Authors' reply (as shown) does not provide the requested concrete evidence; the reviewer remains unconvinced. This weakens both the efficiency claim and the “optimality” argument around Gaussian count.) Reviewer VL78 explicitly states that without "optimized isotropy," the conclusion is not supported and asks to either add the ablation or weaken the claim. The discussion excerpt shows no follow-up from authors providing this ablation. This is a clear example where the rebuttal did not complete the requested validation.

**Reviewer Scores:**

I see the reviewers are not convinced even after the rebuttal.

---

### Decision · Program_Chairs · 2026-01-26

Reject